# A Comprehensive Computational Insight into the PD-L1 Binding to PD-1 and Small Molecules

**DOI:** 10.3390/ph17030316

**Published:** 2024-02-28

**Authors:** Marialuigia Fantacuzzi, Roberto Paciotti, Mariangela Agamennone

**Affiliations:** Department of Pharmacy, University “G. d’Annunzio” of Chieti-Pescara, Via Dei Vestini, 31, 66100 Chieti, Italy; r.paciotti@unich.it

**Keywords:** cancer immunotherapy, PD-1, PD-L1, computational studies, docking, molecular dynamics, virtual screening

## Abstract

Immunotherapy has marked a revolution in cancer therapy. The most extensively studied target in this field is represented by the protein–protein interaction between PD-1 and its ligand, PD-L1. The promising results obtained with the clinical use of monoclonal antibodies (mAbs) directed against both PD-1 and PD-L1 have prompted the search for small-molecule binders capable of disrupting the protein–protein contact and overcoming the limitations presented by mAbs. The disclosure of the first X-ray complexes of PD-L1 with BMS ligands showed the protein in dimeric form, with the ligand in a symmetrical hydrophobic tunnel. These findings paved the way for the discovery of new ligands. To this end, and to understand the binding mechanism of small molecules to PD-L1 along with the dimerization process, many structure-based computational studies have been applied. In the present review, we examined the most relevant articles presenting computational analyses aimed at elucidating the binding mechanism of PD-L1 with PD-1 and small molecule ligands. Additionally, virtual screening studies that identified validated PD-L1 ligands were included. The relevance of the reported studies highlights the increasingly prominent role that these techniques can play in chemical biology and drug discovery.

## 1. Introduction

The connection between cancer and the immune system was suggested for the first time in 1863 when Virchow noticed the infiltration of leukocytes in cancer tissue [1]. Some years later, Coley administered a mix of bacteria, the “Coley’s toxin”, to treat inoperable tumors, obtaining, with a variable clinical response, a substantial reduction of tumor dimensions [2]. In 1971, Burnet and Thomas hypothesized that the immune system is able to control cancer development, recognizing and eliminating tumor cells [3]. Nowadays, the role of the immune system in cancer progression control is widely recognized. In particular, the interplay between the immune system and cancer cells has been better defined with the so defined immunoediting process: elimination, equilibrium, and escape [4].

### 1.1. Immune Checkpoints, PD-1, and Its Binders

The immune response, in particular T-cell activity, is regulated by a complex network of events and involves several actors [5], but a key role is played by immune checkpoints (ICs) that can have a co-inhibitory or co-stimulatory action. In physiological conditions, ICs are responsible for immune tolerance, avoiding autoimmune reactions and tissue damage due to prolonged inflammation [6]. Cancer tissue exploits this mechanism to prevent the immune system from eliminating cancer cells by silencing T-cells. Ipilimumab was approved by the FDA in 2011 as the first example of mAbs used in cancer therapy targeting an immune checkpoint (CTLA-4) and represents a revolution in cancer treatment. The disclosure of the first immunotherapeutic drug paved the way for the research of other tools targeting different ICs, also to overcome the severe immune-related side effects presented by ipilimumab [7]. Since then, the number of clinical trials in the immuno–oncology field has increased almost exponentially [8]. In this context, the most explored targets are PD-1 and its ligands PD-L1.

The programmed death protein PD-1 was identified for the first time by Ishida and coworkers in 1992 [9], even if its effective role was clarified later [10]. PD-1 belongs to the CD28 family and is encoded by the Pdcd1 gene on chromosome 2 (2q37) [11]. It is a glycoprotein expressed mainly on the surface of T- and B-cells, but also on myeloid cells, thymocytes, natural killer (NK) cells, dendritic cells, and monocytes, and its expression is promoted by T-cell activation.

PD-1 binds two endogenous ligands, PD-L1 and PD-L2, identified in 1999 and 2001, respectively, and encoded on the same chromosome 9p24.2 [12,13].

PD-L1 (B7-H1, CD274) is constitutively expressed on antigen-presenting cells (APC) but can be widely located on hematopoietic cells (B-cells, T-cells, monocytes, and dendritic cells), and peripheral non-hematopoietic tissues such as the heart, kidney, lung, placenta, and liver. PD-L1 expression is induced by several pro-inflammatory cytokines (e.g., INF-g, TNF-a, VEGF, and others). PD-1 is not the unique binder of PD-L1 that can interact also with CD-80 [14].

PD-L2 (B7-DC, CD273), the second identified PD-1 binder, has a similar profile to PD-L1 in terms of expression and function, even though it shares a limited identity with PD-L1 (almost 40%). Moreover, PD-L1 and PD-L2 present a sequence identity of 20% with B7-1 and B7-2 that bind CD28 and CTLA-4, respectively.

PD-L2 binds PD-1 differently and with stronger affinity than PD-L1 [15,16] thanks to the PD-1 adaptability and flexibility [17]. However, it is a less-explored target with respect to PD-L1 because of its limited expression [15]. Both ligands, along with PD-1, can be released in soluble form and represent negative prognostic markers in several tumors [18].

### 1.2. The PD-1/PD-L1 Pathway

In physiological conditions, PD-1 expression on the T-cell surface is promoted by T-cell activation. Its expression is accompanied by the release of interferon that fosters the expression of PD-L1 on APC or surrounding tissues. The PD-1 binding to endogenous ligands, PD-L1 and PD-L2, at the immunological synapse strongly inhibits TCR signal transduction and CD28/CD80 co-stimulation (Figure 1). In particular, PD-1/PD-L1(2) contact causes the phosphorylation of the Immunoreceptor Tyrosine-Based Inhibitory Motif (ITIM) and the Immunoreceptor Inhibitory Tyrosine-Based Switch Motif (ITSM) located at the intracellular PD-1 tail. Src homology 2 domain-containing protein tyrosine phosphatase 1 and 2 (SHP-1 and SHP-2) are then recruited by ITIM and ITSM and block the TCR signal transduction [19]. Another effect of the PD-1 binding to its ligands is the PTEN-mediated blocking of T-cell proliferation [20].

The PD-1/PD-L1 interaction results in the reduction of the synthesis of cytokines, the blocking of the activation, proliferation, and acquisition of the effector capacities of the T-cells. In addition, activated PD-1 leads to a suppression of the consumption of oxygen. The oxidation of fatty acids, and no longer aerobic glycolysis, is used as the dominant energy source. Consequently, T-cells with activated PD-1 increase the production of reactive oxygen species helping to create an oxidative environment [21].

As already mentioned, cancer tissue can exploit this physiological mechanism to create an immunosuppressive environment favorable to tumor growth and progression by overexpressing PD-L1 in the escape phase. There are two main mechanisms of PD-L1 up-regulation: the innate immune resistance produced by oncogene suppression, and the adaptive immune resistance that exploits the INF-γ to induce the PD-L1 expression [22]. In both cases, the final result is the immune system silencing [23].

This evidence prompted the search for agents able to interfere with the PD-1/PD-L1 contact to be exploited in cancer treatment. The emergence of mAbs interacting with ICs allowed treating chemo-resistant tumors with surprising efficacy.

### 1.3. Current Drugs

As already mentioned, Ipilimumab was the first example of an immune checkpoint inhibitor (ICI) in clinical use and represented a breakthrough in cancer treatment, despite its severe side effects. Nowadays, the largely most exploited drugs in the immuno-oncology field are human or humanized mAbs targeting PD-1 or PD-L1. A panel of currently approved mAbs by the FDA is reported in Table 1. Apart from those that are available, the investigation of these compounds is still ongoing, with the number of enrolling clinical studies constantly increasing since 2014 [24].

MAbs, in fact, represented a revolution in cancer treatment, and a tool providing a long-lasting resolution for drug-resistant and metastatic tumors. Nevertheless, ICIs have a series of limitations. Targeting PD-1 and PD-L1 can produce immune-related adverse events that can hamper the patient’s treatment. Moreover, depending on the tumor type, just a limited percentage of patients (10–60%) is respondent to the therapy [25]. In this respect, combination therapies could help to reduce side effects and increase the number of patients with beneficial effects [26].

Apart from these aspects, mAbs present several limitations in terms of high production costs, side effects, missing oral bioavailability that forces intravenous administration, prolonged tissue retention, and low membrane permeability. To overcome these shortcomings, research has switched to the development of small molecule inhibitors.

A recent report analyzing clinical trials in the immuno–oncology field highlights a slight but substantial inversion of the previous trends where the number of Phase II trials involving the axis PD-1/PD-L1 is diminishing. The authors highlight the increased interest in other targets involved in cancer immunotherapy, the orientation toward different technologies, and the increased number of combination therapies under study to overcome ICI limits [8,27].

### 1.4. Small Molecule Binders of PD-L1

The identification of small molecules interfering with the PD-1/PD-L1 protein–protein interaction has lagged behind the discovery of the role played by these two important immune checkpoints. The reason for the difficult identification of effective PD-1/PD-L1 small molecule inhibitors can be attributed to the characteristics of the interacting surfaces of the partner proteins that are very flexible and flat, and so hardly druggable.

The first small molecules binding PD-L1 were disclosed in 2015 in two patents from the Bristol Mayer Squibb [28,29], and represent the paradigm compounds as PD-L1 inhibitors until today. Since then, the number of published PD-L1 ligands has increased constantly.

The most important contribution to the discovery of small molecule binders of PD-L1 field has been provided by Holak and coworkers, who investigated the BMS ligands binding mode on PD-L1. The Polish research group resolved the X-ray complexes of the PD-L1 extracellular domain with several BMS compounds. The crystallographic data revealed a ligand/protein ratio of 1:2, with PD-L1 binding the small molecule in dimeric form [30].

The availability of PD-L1 X-ray complexes with small molecules paved the way for the discovery of many other small molecules able to produce the same dimerization process.

Thus, the design strategy shifted from the search for a protein–protein interaction (PD-1/PD-L1) inhibitor to the identification of protein–protein (PD-L1 dimer) structure stabilizers, a new and fascinating field of research [31].

All ligands producing PD-L1 dimerization share a common structural feature consisting in a biphenyl/biaryl portion linked to another aromatic system. Skalniak et al. demonstrated, by NMR studies, that starting from BMS-1166, the minimum fragment-conserving activity is the biphenyl group (Figure 2) [32].

Many reviews published so far have focused on PD-L1 ligands [33,34,35,36,37,38,39]. A very recent review revised the application of computational methods to the discovery of PD-1/PD-L1 inhibitors [40].

The present work explores the computational approaches used to improve the understanding of the binding processes involving PD-L1. In particular, we focused on computational studies aimed at mapping the binding mechanism of PD-L1 with its natural binder PD-1 and small molecules. In addition, the application of in silico methods to the identification of new compounds through virtual screening campaigns was explored. Articles reporting computational studies, in particular docking, as ancillary to other investigations were not considered.

Computational approaches are now an integral part of drug discovery and chemical biology studies. Recent technological and scientific advances have promoted the role of computational methods as leading tools in the prediction of protein structure and function [41] and in the overall drug discovery process, also because of the application of machine learning approaches along with physics-based methods that can exploit increasingly powerful computing systems [42].

## 2. Structural Depiction of PD-L1 and Its Binders

Most of the computational methods presented are based on structure-based approaches. The availability of X-ray data of PD-L1 bound to its endogenous ligand and to small molecules, indeed, fed a plethora of studies exploiting docking, molecular dynamics (MD) and other target-based in silico analyses to understand PD-L1 interaction mechanism with its binders. Here, we review the structural features of the studied proteins, focusing on the available experimental data.

PD-L1 is a 290 amino acids (aa) protein belonging to the type I transmembrane protein family. It is composed of extracellular Ig-V- and Ig-C-like domains, a transmembrane portion, and a short intracellular tail of 30-aa (Figure 3B). The distal Ig-type V-like domain is responsible for the interaction with PD-1 and small molecules.

PD-1 is a 55 kDa type I transmembrane protein of the immune globulin superfamily, composed of an N-terminal Immunoglobulin Variable (Ig-V) extracellular domain, a transmembrane domain, and a cytoplasmic tail (Figure 3A). It is composed of 288 aa and shares the 21–33% sequence identity with CTLA-4, ICOS, and CD28.

The extracellular domain is responsible for the interactions with its ligands, while the intracellular tail has two phosphorylation sites, ITIM and ITSM, essential for its activity.

### 2.1. PD-L1 Structure in the Apo Form and in Complex with PD-1

The first crystallographic data of the hPD-1/mPD-L1 complex (PDB ID 3BIK) were published in 2008 [43], while the fully human complex (PDB ID 4ZQK) was published in 2015 [44]. A list of the PDB data of apo PD-L1 or PD-1/PD-L1 is provided in Table 2.

The contact surfaces between the Ig-V-like domains of hPD-1 and hPD-L1 are arranged orthogonally as a 1:1 complex. The proteins undergo a conformational change during complex formation. This is more evident for PD-1, where the loop CC’ adopts a closed conformation. The interaction area is large (1870 Å^2^) and flat, and hydrophobic and polar interactions take place between PD-1 and PD-L1. (Figure 4).

An insightful analysis of PD-1/PD-L1 structures, complexed with several binders, has been recently published by Boisgerault and Bertrand [45].

**Table 2 pharmaceuticals-17-00316-t002:** Available X-ray structures of apo PD-L1 and PD-1/PD-L1 complex.

PDB ID	Protein	Resolution (Å)	Release	Reference
3BIS	Apo PD-L1	2.64	2008	[43]
4Z18	Apo PD-L1	1.95	2015	Fedorov, A.A., To be published
5C3T	PD-L1 binding domain	1.80	2015	[44]
5JDR	Apo PD-L1	2.70	2017	[46]
6NP9	Apo mutant PD-L1 (V76T)	1.27	2019	[47]
3FN3	Dimeric structure of PD-L1	2.70	2009	[48]
6L8R	Membrane-bound cytoplasmatic domain PD-L1	NMR	2020	[49]
7DCV	Transmembrane domain PD-L1	NMR	2022	[50]
3BIK	Complex mPD-1/hPD-L1	2.65	2008	[43]
3SBW	Complex mPD-1/hPD-L1	2.28	2011	Lazar-Molnar, To be published
4ZQK	Complex hPD-1/hPD-L1	2.45	2015	[44]
5IUS	Complex with high affinity mutated PD-1	2.89	2016	[51]

### 2.2. PD-L1 in Complex with Small Molecule Binders

In 2016, Holak and coworkers deposited the first X-ray data of complexes between PD-L1 and the small molecules BMS-202 and BMS-8 disclosed in the BMS patents (PDB IDs 5J89 and 5J8O, respectively) [30]. The stoichiometry of the complex is 2:1, with the ligand bound at the dimer interface (Figure 5A). The central part of the PD-1 interacting surface represents the small molecule binding site, and the displacement of Tyr56, Met115, and Tyr123 creates a cylindrical hydrophobic cavity that can accommodate the biphenyl group of BMS molecules. In Figure 5B, the superposition of the structures of PD-L1 bound to PD-1 and BMS-202 is shown, while in Figure 5C, the interacting residues of each PD-L1 protein and the contact surface with the ligands are shown.

A complete list of available experimental structures of PD-L1 in the complex, with small molecules reported in Table 3 and X-ray ligand structures represented in Figure 6.

## 3. Computational Studies Contributing to the Binding Mechanism Comprehension

Molecular dynamics (MD) is a powerful computational approach that allows for the exploration of the conformational space of large biological systems.

More specifically, in a classical MD protocol, it is possible to calculate the force applied to each atom in the system and then use Newton’s laws of motion to predict the spatial position of each atom as a function of time [62]. The resulting trajectory describes the conformational change of the 3D structure during the simulation time. The length of a trajectory is, therefore, a critical point of the simulation.

Forces in an MD simulation are calculated using a molecular mechanics force field (FF), which contains structural parameters (for example, the length of each covalent bond), as well as a set of equations to define electrostatic interactions between atoms and other types of interatomic non-covalent interactions [62]. The type of FF adopted for MD simulation can significantly influence the computational results and, therefore, should be selected carefully.

MD is widely applied in structural biology and ligand–receptor interaction studies because it can provide important insight into protein flexibility and inter-domain interactions that can be difficult to study by experimental approaches [63]. MD simulations are often performed starting from X-ray and NMR data to refine the experimental structure and sample its configuration space in order to evaluate the energy changes induced by several stimuli such as mutations, pH, binding with small molecules, etc.

This important computational method was, therefore, extensively applied also to study PD-1, PD-L1, and their complex. Indeed, classical and accelerated MD calculations using the AMBERff14SB force field (FF) were performed [64] to investigate the conformational space of the 19–127 domain of the apo-PD-L1, which is the region involved in the binding with other proteins. The extended MD trajectories (1 μs) and principal component analysis (PCA) provided detailed information on structural displacements in apo-PD-L1, mainly associated with the movement of a specific region (C”D loop), suggesting that the PD-L1 binding process occurs basically by a conformational selection mechanism.

In the next paragraph, we reported some of the most relevant MD studies, often coupled with other computational methods, aimed to characterize the binding mechanism of hPD-L1 to PD-1, an essential step for the design of small molecules able to inhibit this immune checkpoint.

### 3.1. PD-1 Binding

To design effective small molecules able to disrupt the PD-1/PD-L1 pathway, it is crucial to know the 3D structure of the corresponding protein–protein complex. As already mentioned, the X-ray structure of the human PD-1/PD-L1 adduct (PDB ID 4ZQK) was reported for the first time by Holak and coworkers along with the apo-PD-1 binding domain from hPD-L1 (PDB ID 5C3T) [44].

Analysis of the crystal structure of the PD-1/PD-L1 complex (Figure 7A) revealed that the two subunits establish both hydrophobic and polar interactions, with a central hydrophobic core formed by non-polar residues of both PD-1 (Val64, Ile126, Leu128, Ala132, Ile134) and PD-L1 (Ile54, Tyr56, Met115, Ala121, Tyr123) units. The protein–protein interface is also characterized by a buried π–π stacking interaction between the aromatic moieties of Tyr68 (PD-1) and Tyr123 (PD-L1). The interaction region exposed to the solvent is instead characterized by hydrogen bonds and ionic interactions such as Ala132-Gln66 (PD-L1), Ile134-Tyr56 (PD-L1) and Glu136-Arg113 (PD-L1). Other important contacts include Thr76-Tyr123 (PD-L1), Gln75-Arg125 (PD-L1), Thr76-Lys124 (PD-L1), Lys78-Phe19 (PD-L1).

Three main hot regions have been identified on hPD-L1. The first region is a hydrophobic pocket composed of the side chains of Tyr56, Glu58, Arg113, Met115, and Tyr123, which can fit Ile134 of PD-1. The second hot region, located near the previous hydrophobic pocket, is determined by Met115, Ala121, and Tyr123, and accommodates Ile126 of PD-1. The third hot region is formed by the charged side chains of Asp122-Arg125 and Asp26, representing a polar groove that can fit Tyr68, Gln75, and Thr76 of PD-1 (Table 4).

The X-ray data revealed additional structural details. Specifically, the formation of the PD-1/PD-L1 complex requires significant structural flexibility of the hPD-1 unit. Upon superimposing the apo-hPD-1 with the hPD-1 structure extracted from the previously described PD-1/PDL1 (PDB ID: 4ZQK), a high conformational similarity was observed, except for the Met70-Asp77 loop (CC’ loop), which shows an open conformation in the apo-PD-1 structure (Figure 7B). Conversely, the Met70-Asp77 loop undergoes a closed conformation around hPD-L1 in the complex formation (Figure 7A,B), promoting the formation of multiple hydrogen bonds between the two partner proteins. This indicates that the rearrangement of the CC’ loop stabilizes the initial transient interaction between hPD-1 and hPD-L1.

The role of the CC’ loop in the PD-1/PD-L1 complex formation was further investigated by performing MD simulations of apo-PD-1 and the PD-1/PD-L1 complex using the CHARMM36 FF by Liu et al. [65]. The MD results and cluster analysis suggest that the open and closed conformations coexist within a dynamic ensemble in apo-PD-1. The energy barrier is represented by the H-bond pattern that must be broken to allow for conformational repositioning. The PD-1 receptor, in its open conformation, binds to PD-L1 to form the initial encounter complex. This complex undergoes structural rearrangements, resulting in the final closed complex, as observed in the PD-1/PD-L1 crystal structure (PDB ID: 4ZQK). These findings suggest a complex binding process between PD-1 and PD-L1, involving both conformational selection and induced-fit mechanisms.

The PD-1/PD-L1 adduct was also investigated by Kenn et al. [66] performing MD simulations (three replicas of 600 ns) using the Amber99sb-ildn FF coupling with an unsupervised clustering method. They identified specific regions of the PD-1/PD-L1 complex forming stable clusters over time, which are hence named “semi-rigid domains”.

Further insight into interactions between PD-L1 and PD-1 (and different monoclonal antibodies) was provided by Shi et al. [67], adopting an elaborated multi-layered computational approach based on MD simulations (Amber FF14SB FF, simulation time 200 ns), per-residue free energy decomposition, virtual alanine scanning mutagenesis and residue-residue contact analysis. In agreement with other studies previously mentioned, the virtual alanine scanning mutagenesis suggested that Tyr56, Gln66, Met115, Asp122, Tyr123, and Arg125 are the most important residues on the PD-L1 surface for PPI (hot spots). The residue–residue contact analysis further shows that PD-1 interacts with PD-L1 mainly by F and G strands.

The potential hot spots of PD-1/PD-L1 were characterized by using another alanine scanning approach based on single-trajectory MD calculations (MD-based computational alanine scanning) coupled with MM/GBSA/IE method [68]. The results reveal eight hot spot residues for both PD-1 (Gln75, Ile134, Ile126, Glu84, Lys78, Tyr68, Leu128, and Asn66) and PD-L1 (_L_Tyr123, _L_Tyr56, _L_Arg125, _L_Met115, _L_Arg113, _L_Gln66, _L_Ile54 and _L_Lys124). Among them, _L_Tyr123 (PD-L1) was demonstrated to be one of the most significant residues in the PD-1/PD-L1 interaction since it establishes favorable contacts with Ile134, Tyr68, and Glu136 of the PD-1 receptor.

The apo-PD-1 and its complex with PD-L1 were also studied by Du et al. using the MD protocol (OPLS/AA FF, production run 50 ns) combined with computational mutagenesis studies providing crucial information for designing engineered PD-1 mutants to modulate the PD-1/PD-L1 pathway [69]. The MD simulation revealed that not all of the key residues identified in the crystal structure analysis contribute to the protein–protein interaction (PPI) throughout the entire trajectory, indicating their limited involvement in the binding process. Additionally, the solvent-accessible surface area (SASA) calculations showed that the binding surface expands from 220 Å in the X-ray structure to 440 Å after the MD simulations.

The decomposition analysis of the total binding energy computed with MM/PBSA revealed that Arg104, Lys131, and Lys135 are the most important residues on the hPD-1 surface for PPI. Several hPD-1 mutants, including Met70Ile, Ser87Trp, Ala132Leu, and Lys135Met, showed improved hPD-L1 binding ability compared to wild-type hPD-1. These mutants provide important details for modulating the interaction between hPD-1 and hPD-L1.

Another PD-1 mutant with ultra-high affinity for PD-L1 has been obtained and characterized, named high-affinity consensus (HAC) PD-1, showing superior therapeutic efficacy in mice compared with antibodies. The resolution of its X-Ray structure in the complex with PD-L1 (PDB ID 5IUS) showed that HAC PD-1 binds PD-L1, establishing polar interactions [51]. MD simulations (Amber ff14SB FF, 20 ns) revealed that the wild-type PD-1 is affected by a greater conformational variability compared with HAC PD-1 and that the mutations Tyr68His, Met70Glu, Lys78Thr allow the formation of favorable contacts with PD-L1, stabilizing the HAC PD-1/PD-L1 complex.

As previously described, tumor cells overexpress PD-L1 on their surface to elude the immune system. Notably, the tumor microenvironment is generally characterized by acidic pH which affects the protonation states of the residue side chains with consequent effects on PD-1/PD-L1 interactions. Interestingly, Pascolutti et al. found that the HAC PD-1 exhibits pH-dependent affinity, with strong binding at low pH conditions [51]. Indeed, the Tyr68His mutation allows for the formation of salt bridges with the Asp122 of PD-L1 due to protonation of His68 occurring at low pH, greatly improving the stability of the protein–protein adduct.

The effect of acidic pH on the mechanism and kinetics of the HAC PD-1/PD-L1 formation was also investigated by Klyukin et al. [70]. They used the infrequent metadynamics technique (Amber03 FF, production run 100 ns) considering two pH levels, 7.4 and 5.5, corresponding to physiological conditions and acidic tumor microenvironments, respectively [70]. In agreement with Pascolutti et al., their results showed that the PPIs of HAC PD-1/PD-L1 are significantly affected by pH changes, and also that small variations can induce a relevant increase in binding strength. In particular, His68 (PD-1) undergoing protonation at pH 5.5, greatly stabilizes the complex interacting with PD-L1 Asp122.

All the computational outcomes so far described have shed light on the binding mechanism of the PD-1/PD-L1 formation and confirmed the role of hot spot residues identified by Holak. As discussed in the next paragraphs, this crucial information is used for structure-based drug design studies aimed at identifying small molecule inhibitors of the PD-1/PD-L1 pattern.

Table 5 summarizes the main computational methods discussed in the present paragraph.

### 3.2. Small Molecule Binding

As previously stated, the discovery of the initial X-ray complexes of BMS ligands with PD-L1 revealed the dimerization effect caused by small molecules and prompted researchers to use structure-based computational methods to explain the binding and dimerization processes of biphenyl-based inhibitors.

In 2019, Almahmoud and Zhong conducted a study on the binding mode of 29 biphenyl derivatives extracted from the BMS patents through molecular docking studies. The docking calculations aimed to identify the most relevant residues interacting with ligands to design optimized binders. To ensure the consistency of the docking results, calculations were performed on two PD-L1 X-ray structures (PDB ID 5N2F and 5NIU). The computational analysis was conducted using the software packages MOE (Molecular Operating Environment (MOE), Chemical Computing Group ULC, 910-1010 Sherbrooke St. W., Montreal, QC H3A 2R7, Canada) and the Schrödinger Suite (Schrödinger Suite, Schrödinger, LLC, New York, NY, USA). The final results indicate the asymmetric binding of small molecule ligands to the two PD-L1 monomers, which is consistent with the findings of Zak et al. [30]. The most relevant residues are Tyr56 of both chains, which is retrieved in 100% of predicted complexes, followed by _B_Asp122, _B_Lys124, and _B_Arg125 [71].

Sasmal et al. obtained a similar result when investigating the binding site of PD-L1 dimers in the complex with small molecules using more tools. At first, they exploited DoGSiteScorer and Pankweb to quantify the dimensions and properties of the large hydrophobic tunnel formed between the two PD-L1 monomers. Thus, BIOVIA Discovery Studio (BIOVIA, Dassault Systèmes, San Diego: Dassault Systèmes) was used to prepare and dock a comprehensive series of biphenyl derivatives from the literature and patents into the PD-L1 crystal structure (PDB ID 5N2F) to assess their binding mode in the PD-L1 tunnel. The main conclusions are that hydrophobicity is essential for PD-L1 inhibition, the biphenyl system is required to interact with _A_Tyr56, while another aromatic portion is essential for contact with _B_Tyr56 and _B_Asp122. The third aromatic ring must be derivatized by flexible polar chains interacting with the PD-L1 groove [72].

The Alanine-Scanning-Interaction-Entropy (AS-IE) approach was applied to quantify the contribution of single residues to the global binding ΔG of BMS derivatives into the PD-L1 dimer. AS-IE is a computational method developed by Liu et al. [73] that determines the entropic contribution to binding free energy from fluctuations in individual residue–ligand interaction energies in a single MD trajectory. The relative values of ligand binding to the wild type compared to the mutated protein provide the calculated residue-specific binding free energies for each residue. A total of 35 BMS derivatives were divided into five groups based on their inhibition potency and subjected to MD simulation and subsequent AS-IE analysis using AMBER16 with the ff14SB and GAFF force fields.

The residues that contributed the most to the global binding ΔG were _A_Tyr123 and _B_Tyr56, followed by _B_Met155, _A_Met115, _B_Gln66, and _A_Asp122. The analysis suggests that modifications to the third aromatic ring (C) are important for designing new ligands, while substitutions on the A ring of the biphenyl system do not result in better inhibitors [74].

Partial Least Squares Discriminant Analysis and flexible docking studies were carried out by Kuang at al. to elicit features distinguishing between active and inactive inhibitors. The authors collected 2558 PD-L1 inhibitors from the literature and submitted them to the generation of a classification model adding 7674 non-inhibitors randomly selected from PubChem. The classification model, which presented good sensitivity and specificity, reports the most relevant contribution of intramolecular H-bonds, amphotericity, radius of gyration, non-bonded electrostatic energy, octanol−water partition coefficient, and fractional van der Waals surface area of H-bond donors in the discrimination between active and inactive compounds [75].

Shi and coworkers conducted a comprehensive computational study of PD-L1 to gain insight into the dimerization process and to identify the most relevant part of the BMS derivatives that bind to the target protein. They generated an R-group QSAR model that suggested the most relevant substituent position and the residues that mainly contribute to the ligand potency. The results indicate that substituents in the para position of both external aromatic rings have more influence on the ligand potency [76].

In 2019, Mejias and Guirola applied a co-solvent MD simulation to represent a pharmacophore model. The NAMD software with the CHARMM27 force field was used to perform 100 ns MD simulations with three different solvent mixtures (isopropanol/water, acetamide/water, and isopropylamine/acetate) on the PD-L1 monomeric structure (PDB ID 5C3T). The Volmap tool was used to define the occupancy of each probe. Finally, after an energy-based filtering procedure, a final pharmacophore of ten sites was defined, taking into account the properties of probes that interact more strongly with the protein. The obtained pharmacophore was superimposed with known ligands to assess its validity [77].

Similarly, a study from our group used a newly established FMO/GRID-DRY approach for the characterization of polar and hydrophobic interactions between PD-L1 and both PD-1 and BMS ligands. Fragment Molecular Orbital (FMO) is a powerful ab initio method particularly suited for determining the interaction energy between partner proteins or proteins and ligands. It is particularly sensitive to polar contacts, while it is less good at estimating hydrophobic interactions. The coupling with the GRID approach aims to fill this gap by calculating the PD-L1 molecular interaction fields for the DRY probe [78]. The obtained results indicate that the most important residues for hydrophobic contacts are _A_Tyr56, _A_Met115, _A_Ala121, _A_Tyr123, and _B_Ile54, _B_Tyr56, _B_Met115, and _B_Ala121 [79]. In addition, interaction with these residues appears to be responsible for dimerization. On the other hand, polar contacts also play an important role. In particular, those with the so-called G region, are delimited by Asp26, Asp122, Tyr123, Lys124, and Arg125. The results obtained in this work are in good agreement with those presented by Lim et al., who applied FMO calculations and calculated three-dimensional scattered pair interaction energies (3D-SPIEs) between PD-L1 and a series of binders. In particular, the authors examined binding to PD-1, monoclonal antibodies, macrocyclic peptides, and small molecules. After calculating all pair interaction energies (PIEs), only those at a distance of less than 5.4 Å were selected for further analysis and reported in a 3D scatter plot. The results highlighted the presence of a hot spot shared by all types of ligands and formed by Tyr56, Glu58, and Gln66. A second hot spot is characterized by Asp122 and Arg125, which are involved in interactions with PD-1 and mAbs, while small molecules interact mainly with Asp122. A special role is played by Met115, which is centrally located between the two hot spots and interacts with most ligands [80].

Sun et al. investigated the binding of the BMS compound to PD-L1, comparing the 100 ns MD trajectories of the PD-L1/PD-L1 dimer and the same dimer in the complex with BMS-8. The authors demonstrated that the presence of the ligand stabilizes the complex, increasing the number of interacting residues from eleven to thirteen, and the number of salt bridges from four to six. The molecular dynamics simulations confirm the role of key residues identified in the X-ray and in previously mentioned articles. It also highlights the conformational rearrangement of several residues at the dimer interface to host the ligand and improve the interaction network. Additionally, the binding energies of different systems, including PD-1/PD-L1, PD-L1/BMS-8, PD-L1/PD-L1, and PD-L1/PD-L1 in the complex with BMS-8, were calculated to better depict the small molecule binding mechanism. The resulting ΔG values suggest that PD-L1 prefers to bind with BMS-8 over PD-1. Additionally, the initial binding with the small molecule triggers dimerization [81].

Riccio and colleagues investigated the effect of the tumor microenvironment pH on the binding of ligands to PD-L1 in their paper. The tumor microenvironment is known to have a lower extracellular pH compared to normal tissue, which can affect processes influenced by electrostatic interactions such as protein folding and molecular recognition. The authors investigated the effect of lower pH on the binding of PD-L1 with four ligands: a macrocyclic peptide (peptide-57) and three biphenyl derivatives (BMS-202, S7911, and VIS1059). Docking and MD simulations were conducted to determine the most stable electrostatic and hydrogen bond interactions between the ligands and the protein. The pKa values of the ligands were evaluated using both Marvin and Epik. Microscale thermophoresis was used to experimentally assess pH-dependent variations in binding affinity following computational analysis. The results suggest that ligands containing a basic function that interacts with a negatively charged residue (Asp122) can increase their binding affinity at lower pH. These findings provide insight into the design of high-affinity ligands that account for pH-dependent binding [82].

The following Table 6 reports the key PD-L1 residues in the interaction with small molecules, as defined in the considered studies. The main ligand structural features affecting PD-L1 binding are schematized in Figure 8.

### 3.3. Dimerization

Closely connected to the binding of biphenyl derivatives to PD-L1, the target dimerization promoted by small molecules has been investigated by several researchers. The dimerization process needs to be elucidated in view of the potential relationship between the inhibitory activities of BMS small molecule inhibitors and the stability of the dimerized complex systems. In this context, computational approaches can represent an almost unique tool to understand this unpredicted event.

In 2019, Soremekun et al. carried out all-atom MD simulations to study the dimerization process promoted by BMS-1001 and BMS-1166. Three different systems were built: two unbound PD-L1 monomers and one PD-L1 bound to BMS-1001, and BMS-1166, in the presence of the other PD-L1 monomer in the simulation box. MD simulations of 150 ns were carried out using AMBER14 and the FF14SB force field. During the simulation, the ligand in the bound systems transitioned from the starting monomer to the second one in an intermediate phase. The final simulated state of the bound systems was represented by the dimeric form. In contrast, the unbound PD-L1 did not demonstrate any dimerization, indicating that the studied process is promoted by the ligand. Additionally, ligand binding caused an increase in residue fluctuation compared to the unbound protein. The per-residue energy decomposition analysis for both ligands was estimated using MM/PBSA. The analysis reported a high electrostatic contribution for _A_Asp122, _A_Tyr123, _A_Lys124, and _A_Arg125 outside the hydrophobic tunnel and _B_Ala121 and _B_Asp122 inside the tunnel, along with both _A_Tyr56 and _B_Tyr56 hydrophobic contacts [83].

A similar study characterizing the binding and unbinding process of BMS-8 and BMS-1166 to PD-L1 was presented by Shi et al. using MD simulations and metadynamic studies. They used the X-ray complexes with PDB ID 5J8O and 5NIX and conducted canonical MD simulations with the AMBER FF14SB force field, with a production phase of 150 ns. The authors used MM/GBSA and MM/PBSA methods to calculate the global binding free energies and the contribution of each residue through per-residue-based decomposition analysis. Additionally, they conducted metadynamics to describe the unbinding process, defining two collective variables that accounted for the ligand position in the target protein. The authors confirmed the stabilizing effect of both ligands on PD-L1 and their preferential binding to one monomer over the other. BMS-8 induced greater flexibility in the system compared to BMS-1166, which can be attributed to a larger enthalpic contribution. Metadynamics simulations suggest that the dimerization process is caused by the ligand binding to one monomer, which then recruits the second monomer, in agreement with previous predictions. The most likely dissociation mechanism involves the ligand disengaging from the dimer, which then oligomerizes after the ligand leaves [76].

In 2021, Guo et al. investigated the role of ligand chirality in PD-L1 binding. They performed docking, MD simulation, and per-residue-based decomposition analysis to study the binding of (*R*)- and (*S*)-BMS-200, along with a modified version of BMS-200, where the chiral hydroxyl is substituted by a carbonyl function. The PDB complex with BMS-200 (PDB ID 5N2F) was used as the starting point for docking with AutoDock Vina and MD simulation using GROMACS2106.4. The study confirms the dimerization mechanism induced by ligand binding to one monomer, as previously ascertained. Additionally, the authors highlight a slight difference in binding energy between the two enantiomers, with the R enantiomer showing more interactions. The residues primarily involved in ligand contacts are Ile54, Tyr56, Met115, Ala121, and Tyr123 [84]. The same research group applied a similar computational approach to investigate the binding mechanism of BMS-202 and its modified analogues where the terminal carbonyl group is substituted by a hydroxyl function, generating both enantiomers. In addition to previous results, the authors emphasized the role of the conformational rearrangement of _A_Tyr56, _A_Tyr123, and _B_Met115 in the ligand association and dissociation process [85]. A comprehensive computational study was carried out by Ahmed et al. that explored the binding properties of four BMS ligands comprising the minimum fragment identified by Skalniak et al. [32] The authors also studied the PD-L1/PD-1 complex, the PD-L1 dimer without ligands, and the naturally occurring ‘back-to-back’ PD-L1 dimer (PDB ID 5JDR) through MD simulations and MM-GBSA analysis. Grid Inhomogeneous Solvation Theory (GIST) and Hydration Site Analysis (HSA) were applied to understand the role of water displacement in ligand binding and dimerization. In addition, the authors carried out a 2D-QSAR analysis using 403 ligands extracted from literature and designed a large virtual library of potential PD-L1 ligands. The MD simulations allowed for the clarification of the dimerization process, which is in line with the proposed ligand-induced mechanism. The authors confirm the biphenyl system as the minimum structural ligand requirement for PD-L1 dimerization. The computational solvent mapping suggests that BMS ligand to PD-L1 monomer can be favored by the displacement of unfavorable water molecules from their highly energetic hydration site. This result suggests that substitution of the A ring can further contribute to this effect [86].

## 4. Computational Studies Contributing to the Identification of New Compounds

As the final part of this review, we present computational approaches that have been applied to identify new PD-L1 ligands. As previously mentioned, this paragraph focuses on papers referring to virtual screening campaigns, while neglecting docking studies of newly synthesized compounds, which are less attractive from a computational point of view.

Only papers that include a biological assay demonstrating activity towards PD-L1, and confirming the validity of the computational results, were considered for this review. The preferred method for determining the ability of a compound to inhibit PD1/PDL1 interaction is through biological assays, with the well-established HTFR assay being the preferred option. In silico methods can also be used to virtually test a large number of compounds quickly and inexpensively, reducing the number of compounds to be tested in vitro or in vivo, speeding up the process and reducing costs.

Virtual screening (VS) techniques can be divided into two approaches: ligand- and structure-based. The ligand-based approach is useful when the 3D structure of the target is unknown and relies on the knowledge of the chemical properties of active compounds. However, this method may limit the ability to identify compounds with different structures and/or different types of interactions. On the other hand, the structure-based VS can be used when the 3D structure of the target is known. Due to the availability of numerous 3D PD-L1 structures in the Protein Data Bank, structure-based VS has become a popular method for identifying new ligands in recent years. This involves the screening of commercial and/or in-house databases to find new compounds. Several research groups conducted docking-based VS on diverse small molecule databases, including synthetic and natural compounds, as well as approved drugs. They used various methods to filter the databases and employed one or multiple 3D structures of proteins and ligands stored in the Protein Data Bank.

The section below describes the most significant structure-based VS approaches, including the most popular procedures and database filtering strategies.

Wang et al. screened the Specs database that contains more than 200,000 compounds using the 3D structure of dimeric PD-L1 protein in the complex with BMS-202 (PDB ID 5J89). They used Schrödinger programs such as Protein Preparation Wizard to refine protein structure, and LigPrep to process compounds, exploiting Epik at pH = 7.0 to predict ionized states, tautomers, and stereoisomers. The grid box was centered on the crystallographic ligand and the XP protocol was used to perform docking. The top compounds were re-docked using a flexible docking strategy (Induced Fit Docking), and the Canvas module was used to cluster the best-docked compounds and analyze the binding mode. Compound APBC (SPECS No. AG-690/11449006, Figure 9) binds, like BMS-202, to the hydrophobic site of two PD-L1 monomers with similar anchoring residues Tyr56, Met115, and Ala121. A π–π stacking interaction between the aniline group of APBC and Tyr56 was observed, and a key H-bonding to D122 stabilized the complex. The activity of compounds was valued by HTFR PD-1/PD-L1 interaction assay, with an IC_50_ of 27.82 μM [87].

Lung et al. screened the natural product dataset of the ZINC12 database (ZBC, 180,131 chemical structures) using a docking-based VS with the 3D structure of PD-L1 (PDB ID: 5J89) and the iDock program. To validate the screening protocol, four inhibitors (BMS-8, BMS-37, BMS-200, and BMS-202) with known IC_50_ values were added to the database. An arbitrary cutoff iDock score was set at the value of BMS-202 score (−9.95 kcal mol^−1^) for further analysis. Contact fingerprint analysis was performed using the AuPosSOM web server, which automatically analyzes poses using self-organizing maps. A total of 368 compounds were clustered together with the four known BMS compounds and were filtered by drug-like properties using Data Warrior with the following criteria: molecular weight between 55 and 500 Da, no more than 5 hydrogen bond donors, no more than 10 hydrogen bond acceptors, a partition coefficient logP between −1 and 5, a net charge between −2 and 2, a topological polar surface area less than 100, and no risk of mutagenicity, tumorigenicity, irritation, or reproductive effects. The filtered 111 compounds were clustered using the FragFP descriptor with a minimum similarity of 0.8. Finally, 22 compounds with iDock scores better than BMS-202, similar contact fingerprints, and preferable drug properties were selected for in vitro evaluation. ZINC12529904 (Figure 9) inhibited approximately 40% of the PD-1/PD-L1 interaction at 100 nM in the AlphaLISA binding assay [88].

Barnwal et al. used the ParDOCK program to screen small-molecule drugs from the ZINC database, and the best-scored 11 compounds (docking score < −8 Kcal mol^−1^) were subjected to MD simulation using the AMBER suite to determine the dynamic stability of the interaction. Among these compounds, Ponatinib (Figure 9), a tyrosine kinase inhibitor, exhibited stable binding to the active site of PD-L1, mediated by hydrophobic contacts with Glu54, Glu55, Asp56, Gln49, Val51, Tyr39, Ser100, Ile99, Met98, Ala104, and Asp105. A cell-free fluorescence-quenching study confirmed the binding of ponatinib with recombinant PD-L1 (IC_50_ 1.91 μM) [89].

Acúrcio et al. conducted a docking-based VS on several synthetic compound libraries, including NCI, Enamine, Specs, and in-house databases, comprising almost 900,000 small molecules. The crystallographic complex of BMS-202 and PD-L1 (PDB ID 5J89) was prepared and minimized using the MOE (Molecular Operating Environment (MOE), Chemical Computing Group ULC, 910-1010 Sherbrooke St. W., Montreal, QC H3A 2R7, Canada) software package with Amber 10 EHT force field. The GOLD suite was used to analyze the binding conformations, with _A_Tyr56 set as the center of the binding pocket and a 10 Å radius. A preliminary screening was performed using the ChemPLP fitness scoring function and 50 genetic algorithm (GA) runs, while the top 1000 highest-ranking compounds were analyzed in a more detailed molecular docking study (GoldScore fitness scoring function with 500 GA runs). 95 compounds, resulting from filtering the docking score, fitting the active site, interacting with nearby residues, and Lipinski’s rule of five criteria with the FAF-Drugs4 tool, characterized by variable chemical scaffolds, underwent testing for PD-L1 HTRF assay. Even if a total of 16 compounds, tested at 100 μM, were able to reduce the HTFR signal to 50%, the most promising compound was the **69** (IC_50_ 96 nM, Figure 9) [90].

Bianconi et al. performed a structure-based VS of 5801 small molecules (807 internal subset molecules with MW ≤ 500 Da and 4994 highly soluble Life Chemicals fragments with MW ≤ 300 Da) to identify compounds with good activity at pH 6.2, useful for overcoming drug resistance mechanisms due to an acidic tumor microenvironment. Asp122 and Lys124 residues, responsible for pH-dependent binding activity, were identified as hot spots on PD-L1 (PDB ID 5J89) and were used as key features for selecting hit compounds. Docking studies were conducted using the standard precision (SP) method of Glide and the G-score scoring function. LigPrep was used to refine the ligands, Protein Preparation Wizard to process and energetically refine the C/D PD-L1 chains, and the grid box was centered near Asp122, and Lys124 of chain C. Microscale thermophoresis (MST) experiments were used to confirm the binding to PD-L1 of the top 60 compounds based on their G-score value and interaction with the hot spot residues. The most active compound, VIS310 (Figure 9), showed a high-micromolar dissociation constant (Kd = 163.75 ± 33.61 μM), and better binding efficiency index (BEI = 21.0) than BMS-202 (Kd = 8.13 ± 1.38 μM; BEI = 12.1). Since VIS310 was the only substituted benzamidoxime of the analyzed database, the analogue-based approach to screen the REAL database of Enamine, containing about 43.8 million drug-like compounds, was used to define the structure-activity relationships of benzamidoxime compounds [91].

Wang T. et al. filtered the ChEMBL25 database (1.9 million compounds), based on Lipinski’s rule of five and REOS rules. The selected compounds were submitted to structure-based VS using the 3D structure of the PD-L1 dimer co-crystallized with BMS-202 (PDB ID: 5J89) and Autodock 4.2 with a Lamarckian genetic algorithm (LGA). The first run of screening consisted of 1 round of docking while the top 50,000 molecules were docked five times for ranking purposes. From the top 100 molecules, clustered into 20 groups using FCFP_4 fingerprints, nine compounds (one from each cluster) were selected based on the drug-likeness and structural diversity of the PD-1/PD-L1 TR-FRET assay. The compound with the highest activity (IC_50_ 64.11 μM) was analyzed through a docking study in PD-L1 (PDB ID 5J89), revealing a deep insertion into the hydrophobic cleft formed at the dimer interface of PD-L1. To improve hydrophobic contacts, π–π stacking, and alkyl–π interactions with Tyr56 and Tyr123, a series of biphenyl analogues of BMS dimerizers were synthesized and tested. Compound **17** demonstrated the highest activity with an IC_50_ of 27.8 nM. The complex **17**/PD-L1 was crystallized with a resolution of 2.4 Å (PDB ID 7DY7) [57].

Another useful approach to finding new PD-L1 binders is the generation of a pharmacophore model based on the structure of the crystal ligands and using it for database screening. Wang F. et al. carried out cross-docking of four available crystal complexes (PDB ID 5J8O, 5J89, 5N2D, and 5N2F) using Glide SP and XP to select the best performing PD-L1 structure (PDB ID 5J89) based on the best RMSD in Glide docking and the best AUC in XP. Almost 200,000 compounds of the Specs database were pre-processed by PAINS filtration using the comprehensive application of the Molinspration Cheminformatics software, OpenEye, ChemAxon, RDkit, and CERTARA. A total of 120,000 compounds were analyzed by a pharmacophore model using Phase, a module of the Schrödinger Suite, on the co-crystallized inhibitors BMS-8, BMS-37, BMS-200, and BMS-202 (PDB ID 5J8O, 5N2D, 5N2F, and 5J89, respectively). The condition of mapping at least four of the six pharmacophore features of the model (one positive charge, one H-bond acceptor, one hydrophobic, and one aromatic ring feature) was satisfied by 10,125 compounds. SP and XP docking protocols, using PDB ID 5J89, in Glide, based on the ROC curves and AUC values, were used. The top 1100 molecules (XP G-scores ≤ −9.000 kcal mol^−1^) were ranked and clustered in the Canvas module into 402 groups with Tanimoto coefficients of less than 0.5 to ensure maximum structural diversity among compounds. Best-scored compounds of each cluster were submitted to flexible docking (Induced Fit protocol) to refine the final selection. 91 compounds were purchased and tested by SPR and CBPA (SPECS No. AN-465/42833793, Figure 9) exhibited bioactivity at the molecular (Kd 48.10 μM) and cellular levels [92].

Choorakottayil Pushkaran et al. developed a 3D pharmacophore model considering the key interacting residues between BMS-202 and PD-L1 dimer (PDB ID 5J89) using the “Structure-based pharmacophore” module of Ligand Scout 4.1. The pharmacophore was characterized by seven chemical features: four hydrophobic (H), one positive ionizable (P), one H-bond acceptor (A), and one H-bond donor (D). The obtained model was validated by calculating the enrichment factor (EF) using a test dataset composed of 61 known PD-L1 inhibitors and 1425 decoys. The validated 3D pharmacophore model was used to screen all the FDA-approved drugs in the DrugBank database (1925 compounds) and small molecules in the Specs database (540,807 compounds), with the VS module of Ligand Scout software. Compounds passing the pharmacophore selection matching at list five chemical features (12 in DrugBank and 15,276 in Specs) were submitted to High Throughput VS, SP, and XP docking protocols in Glide. Ligands were prepared by LigPrep, and energy minimization was performed using the OPLS2005 force field while retaining the input structure chirality. The protein crystal structure was minimized by Protein Preparation Wizard, and the grid box was centered on BMS-202. Hits were ranked using the XP docking score, and molecular interactions between the hits and the protein were analyzed using PyMol and Biovia Discovery Studio Visualizer. The cutoff of −9 kcal mol^−1^ allowed the selection of three Drug Bank and eight Specs hits characterized by the key interactions with Tyr56, Met115, and Ala121. ADME and drug-likeness prediction were calculated using the QikProp module of Schrödinger. The in vitro toxicity and PD-1/PDL-1 inhibitory activity established that the drugs Raltitrexed, Safinamide, and the natural AK-968/40642641 (Figure 9) could be used as PDL-1 inhibitors [93].

Fattakhova et al. conducted a combined ligand- and structure-based VS to identify small molecules active on PD-L1. For the structure-based screening, ensemble docking using 7 crystal structures (PDB IDs 5N2F, 5NIU, 6R3K, 5J89, 5J8O, 5N2D, 6NM8) with approximately 10,000 approved or investigational drugs, using the AutoDock Vina algorithm, was performed. The docking protocol was validated by redocking cognate ligands and evaluating the RMSD with the crystal ligand. After merging the data of the seven proteins, the top 1000 molecules were visually inspected and twenty compounds mimicking the key ligand-PD-L1 interactions (strong hydrophobic interactions with several amino acids lining the channel-like pocket of the dimer, π–π interactions with key amino acids like Tyr56, and possible hydrogen and halogen bonds at the channel opening) were selected. ROCS 3.4.1.0 was used for ligand-based screening, which evaluates shape-similarity to the 7 crystallographic ligands. A multi-conformer database of approved and investigational drugs was screened and ranked based on the ROCS_TanimotoCombo score. The top 1000 molecules were docked in the high-resolution PD-L1 crystal structure (PDB ID 5N2F), and after combining molecules obtained by the two screening, 25 compounds were subjected to biological assays. Pyrvinium (Figure 9), an FDA-approved anthelmintic drug belonging to the phenylpyrroles class, showed comparable potency in HTRF and AlphaLISA assays, confirming its potential PD1/PD-L1 inhibitory activity (IC_50_ 29.66 μM). A post-docking optimization of the best docking pose, using the default relaxation protocol in the Desmond Molecular Dynamics v3.6 package, demonstrated that the dimethyl-phenylpyrrole moiety occupied the distal end of the PD-L1 dimer pocket similarly to the PD-L1 cognate ligand [94].

A machine learning approach was used by Patil et al. to discover bioactive PD-L1 dimerizers. They developed models based on 2D chemical descriptors using a series of small-molecule PD-L1 ligands patented by BMS. Multiple 2D fingerprint descriptors (FP1, FP2, Layered, MACCS, Morgan, RDKit) implemented in the Open Drug Discovery Toolkit (ODDT), were calculated. These descriptors were fitted by Random Forest models to 1581 “Active” molecules (BMS molecules), 50 “decoy” molecules per active compound (obtained from the DUD-E database), and 417 known inactive molecules. According to the good correlation coefficient (R) implemented in the ODDT, all fingerprints were used to screen the commercial Cayman Chemical database (16,191 bioactive molecules) and 361 compounds emerged as potentially “active” in at least 5 fingerprint models. A structure-based docking study was then realized using the highest resolution X-ray PD-L1 structure (PDB ID 5N2F) with AutoDock Vina. The binding mode of the cognate ligand (8HW) was correctly predicted (docking score −11.4 kcal mol^−1^). The top 20 compounds, tested in the HTRF PD1-PDL1 binding assay, were selected based on the presence of typical interactions between PD-L1 and its inhibitors and their orientation relative to 8HW. MD simulations were used to predict the binding stability of the three most active compounds using Desmond, with a simulation time of 5 ns. The compound CRT5 (IC_50_ 22.35 μM, Figure 9) is the most active and stable and binds similarly to the crystal ligand. Compound P053 (IC_50_ 33.65 μM) follows a similar trend [95].

In Table 7, the main characteristics of the virtual screening campaigns reported in this paragraph are summarized.

## 5. Application of AI-Based Methods to the Study of Immune Checkpoint Inhibition

Artificial intelligence (AI) has had a major impact on technologies and all fields of science, including structural biology and in silico drug discovery studies. Apart from the already discussed VS campaign by Patil et al. [95], not many other studies report the application of these approaches to the discovery of PD-L1 ligands. Most of the literature mentioning PD-L1 and AI is based on the application of these methods to aid in the diagnosis and prediction of responses to AI treatments [96].

AI is behind a breakthrough in structural biology represented by AlphaFold and similar software (RosettaFold, OpenFold, and ESMFold), which is able to predict the 3D structure of a protein from its amino acid sequence [41,97]. This specific computational method was used to predict the anti-PD-L1 antibody and antigen structures of the humanized 3D5 antibody, named h3D5-hIgG1, as well as the structure of the corresponding h3D5/PD-L1 complex [98]. Experimental results showed that h3D5-hIgG1 is characterized by an extraordinary binding affinity to the PD-L1 protein compared to the parental murine 3D5 antibody. Other Alphafold applications aimed at identifying PD-L1 inhibitors have recently been reviewed by Sobral et al. [40].

The power of AI in drug discovery has also been applied to other targets in the immune checkpoint inhibitor space, as shown in a recent study using an AI algorithm based on deep convolutional neural networks to identify small-molecule inhibitors of cytotoxic T-lymphocyte-associated protein 4 (CTLA-4) that disrupt the CTLA-4/CD80 interaction [99]. This AI approach was used to virtually screen a library of 10 million compounds. The most promising compounds were evaluated using biochemical, biophysical, immunological, and animal assays to demonstrate their ability to inhibit the CTLA-4/CD80 pathway.

Although the application of AI-based methods to the identification of PD-L1 ligands has not been widely exploited, the increasing number of available tools and technical capabilities will certainly encourage the application of these methods to this target.

## 6. Conclusions

The interaction between PD-1 and PD-L1 represents a well-established and valuable target for cancer immunotherapy. Several mAbs are currently in clinical use and represent a new paradigm in cancer therapy.

This protein–protein contact has been extensively investigated using experimental and computational approaches due to its significance. Our review focuses on the application of computational methods to explore the binding mechanism of PD-L1 to PD-1 and small molecule binders. The latter may represent an important goal in the field of immunotherapy, as low molecular weight compounds can overcome limitations due to mAbs administration and side effects. In this context, the resolution of the first X-ray complexes between PD-L1 and BMS derivatives has paved the way for the disclosure of a series of compounds capable of promoting the same dimerization induced by biphenyl compounds.

Most investigations in this field use the available X-ray data to perform docking, MD simulations, and other structure-based approaches. The studies presented here largely confirm the involvement of PD-L1 residues in both the binding of BMS derivatives and the dimerization process, as observed experimentally by Holak. The identification of hot spot residues in the PD-L1 binding region for both PD-1 and small molecules converged on hydrophobic contacts with Tyr56, Met115, Ala121, and Tyr123, as well as electrostatic contacts with Asp122, Lys124, and Arg125. Most of the computational studies completed the binding profile of the studied ligands; therefore, since they mostly overlap with experimental data, it is not trivial to elicit the effective contribution of these methods to the identification of new small-molecule binders.

Several MD simulations, along with other approaches such as MM/GBSA, MM/PBSA, per-residue energy decomposition, and ab initio FMO calculations, were used to gain insight into the dimerization process. The results indicate that the small-molecule ligand binds preferentially to one monomer. This binding is favored by the displacement of “unhappy” water molecules and electrostatic interactions with polar residues in the groove region. This initial contact then recruits the second PD-L1 monomer. It was found that dimerization does not occur in the absence of a ligand and that the biphenyl group (or a bi-aromatic moiety) is the minimum structural requirement for the ligand to promote this process.

The availability of X-ray complexes and activity data for biphenyl ligands, which have fed structure-based virtual screening campaigns, has a twofold implication: while it has provided an essential starting point for the development of other ligands, it has also placed a constraint on the identification of molecules with different scaffolds that can bind PD-L1 by a mechanism of action other than dimerization (assuming that compounds with this activity can exist, given the properties of PD-L1, whose surface druggable sites are unlikely to be identified). Thus, the question of the possible identification of other ligands with a novel mechanism of action is still an open one.

In addition to the need to find a good ligand, one of the most challenging aspects of ICI is the limited number of patients who respond to therapy. Better profiling of cancer protein expression and the adoption of combination therapies may increase the number of patients who can benefit from immunotherapy [100]. Recently, a series of in silico methods was applied to simulate the different intracellular signaling affecting the PD-1/PD-L1 pathway in neuroblastoma (NBM) [101]. In particular, they developed a specific network of protein kinase cascades where the corresponding Michaelis–Menten kinetics parameters were used to create a system of ordinary differential equations. The resulting computational model represents an interesting tool to predict the relation between NBM tumor phenotype and the response of anti-PD-1/PD-L1 therapy, as well as to manage the immunotherapeutic treatment of NBM patients. The latter represents an example of the even more relevant role that computational studies can play in the valuable field of immune checkpoint modulation.

## Figures and Tables

**Figure 1 pharmaceuticals-17-00316-f001:**
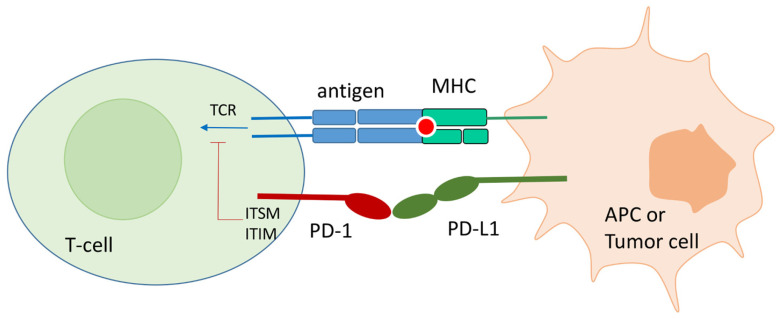
Schematic representation of the PD-1/PD-l1 contact in physiological conditions, and in tumor environment. Monoclonal antibodies directed toward PD-1 and PD-L1 can block the protein–protein contact and reactivate T-cell function.

**Figure 2 pharmaceuticals-17-00316-f002:**
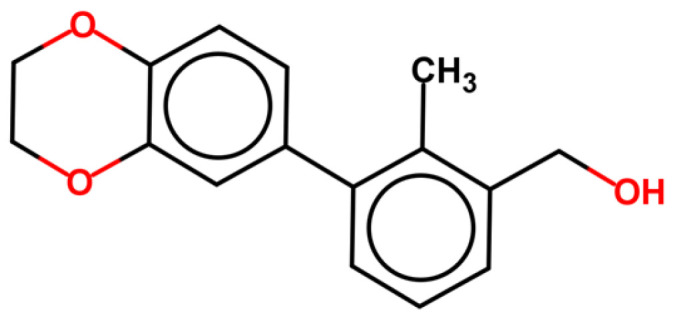
BMS-1166-derived fragment maintaining activity toward PD-L1 in ^1^H-^15^N HMQC NMR assay [32].

**Figure 3 pharmaceuticals-17-00316-f003:**
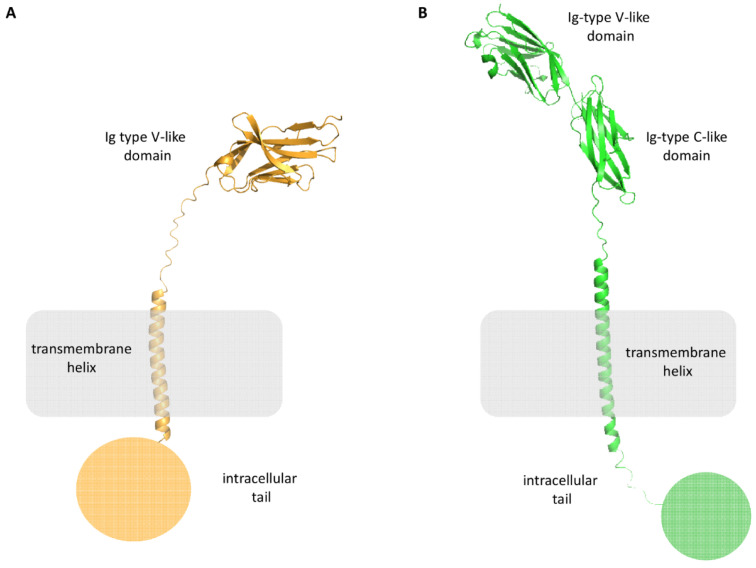
Structural representation of PD-1 (panel (**A**), orange cartoon), and PD-L1 (panel (**B**), green cartoon).

**Figure 4 pharmaceuticals-17-00316-f004:**
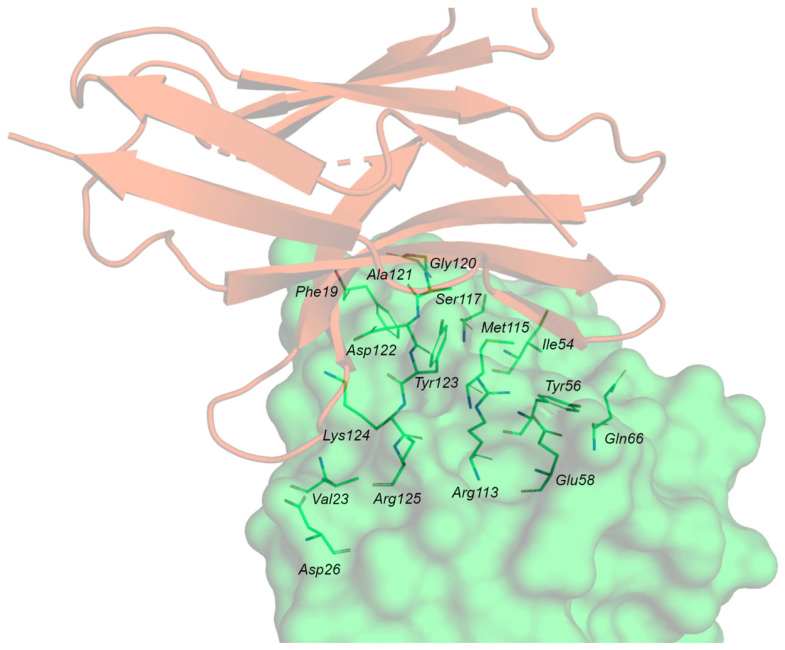
PD-1/PD-L1 complex. PD-1 is represented by the orange cartoon, and PD-L1 as the green surface with the contact residues in green sticks.

**Figure 5 pharmaceuticals-17-00316-f005:**
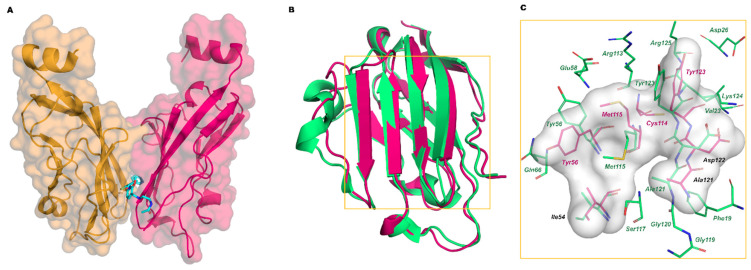
(**A**) Complex of BMS-202 (cyan stick) and PD-L1 dimer (chain A hot pink, chain B orange mixed carton/surface representation); (**B**) Superimposition of the crystal structure of PD-L1 bound to PD-1 (green carton, PDB ID 4ZQK) with PD-L1 bound to BMS-202 (hot-pink carton, PDB ID 5J89); (**C**) Zoom on the interaction residues of PD-L1/PD-1 (green) and of PD-L1/BMS-202 (hot pink). The interaction surface of PD-L1/BMS-202 is represented in grey.

**Figure 6 pharmaceuticals-17-00316-f006:**
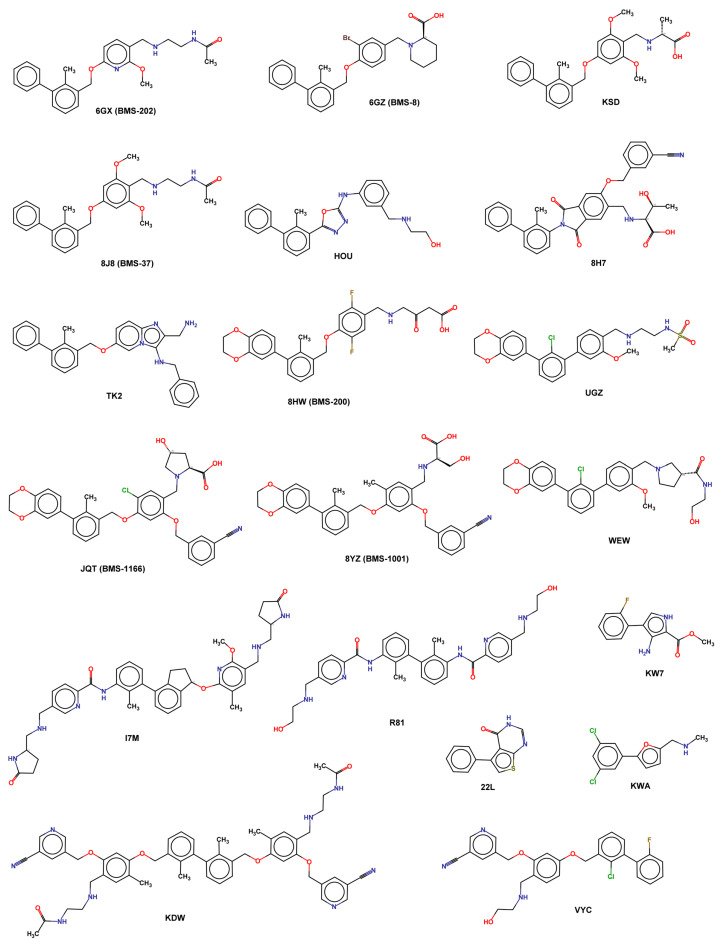
2D structure of X-ray ligands.

**Figure 7 pharmaceuticals-17-00316-f007:**
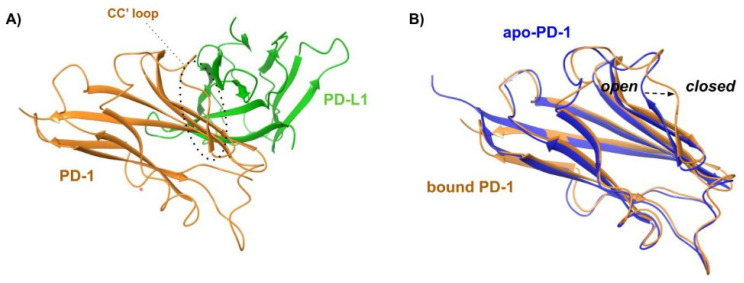
(**A**) Cartoon representation of the X-ray structure of the complex between human PD-1 (orange cartoon) and PD-L1 (green cartoon) resolved by Holak and coworkers (PDB ID 4ZQK). The CC’ loop of PD-1 in its closed conformation is surrounded by a dashed line. (**B**) Superposition of X-ray structures of the bound conformation of PD-1 (orange profile, PDB ID 4ZQK) and apo-PD-1 (blue profile, PDB ID 5C3T). The rearrangement from the open to the closed conformation, which occurs moving from the apo to the bound state, is marked by a black arrow.

**Figure 8 pharmaceuticals-17-00316-f008:**
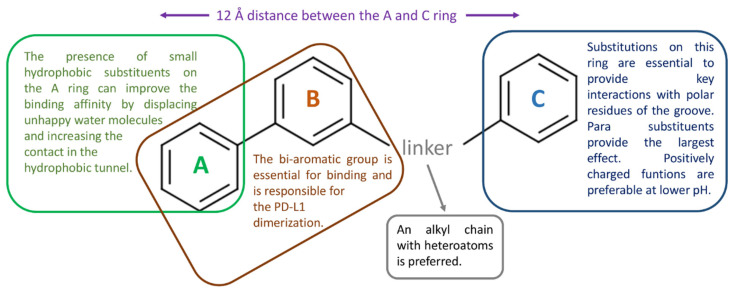
Schematic representation of the main structural features of the bi-aryl PD-L1 binders that are more likely to affect target binding, as suggested by the reviewed articles.

**Figure 9 pharmaceuticals-17-00316-f009:**
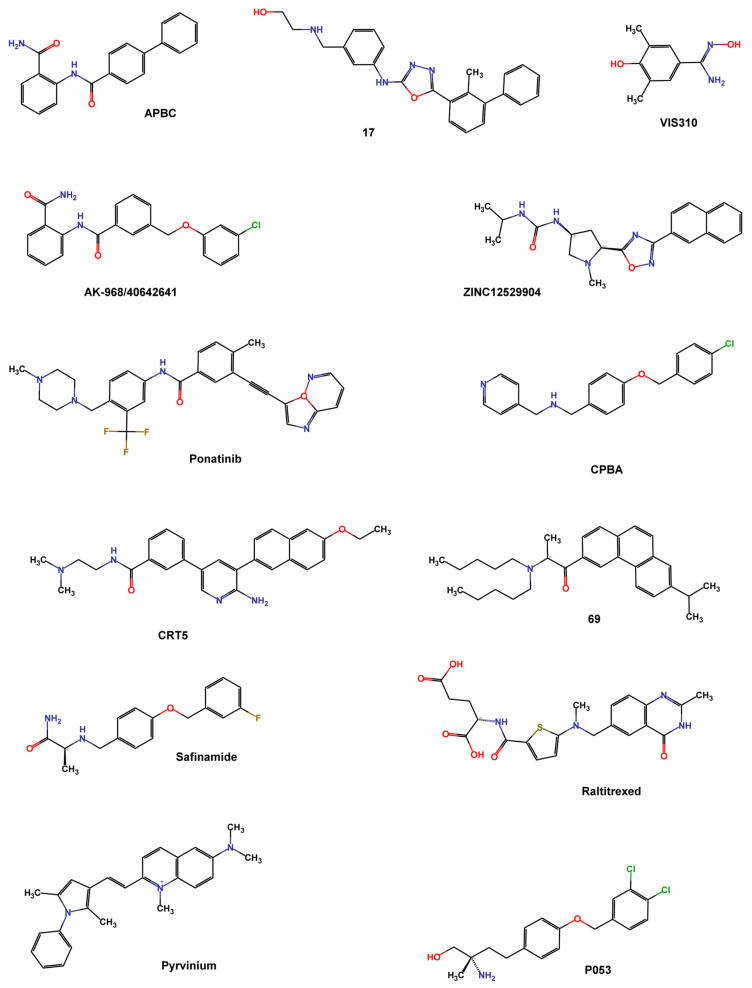
Chemical structures of PD-L1 inhibitors identified by structure-based virtual screening and machine learning.

**Table 1 pharmaceuticals-17-00316-t001:** The FDA approved anti-PD1 or PD-L1 mAbs reported in the Drug Bank (https://go.drugbank.com/, accession date 22 January 2024).

DrugBank ID	Name	Year of Approval	Target	Commercializing Company
DB09035	Nivolumab	2014	PD-1	BMS
DB09037	Pembrolizumab	2014	PD-1	Merk
DB14707	Cemiplimab	2019	PD-1	Sanofi
DB15627	Dostarlimab	2021	PD-1	GSK
DB15766	Retifanlimab	2023	PD-1	Incyte Biosciences
DB11595	Atezolizumab	2016	PD-L1	Genentech
DB11945	Avelumab	2017	PD-L1	Merk
DB11714	Durvalumab	2017	PD-L1	Astra Zeneca

**Table 3 pharmaceuticals-17-00316-t003:** PDB data of crystal of the complex small molecule/PD-L1.

PDB ID	Resolution (Å)	Ligand	IC_50_ Value (nM)	Reference
5J89	2.20	6GX (BMS-202)	18	[30]
5J8O	2.30	6GZ (BMS-8)	146	[30]
5N2D	2.35	8J8 (BMS-37)	6–100	[52]
5N2F	1.70	8HW (BMS-200)	80.00	[52]
5NIU	2.01	8YZ (BMS-1001)	2.25	[32]
6NM7	2.43	22L	n.d.	[47]
6NM8	2.79	KSD	53.00	[47]
6NOJ	2.33	KW7	Kd = 1.9 mM	[47]
6NOS	2.70	KWA	Kd = 1.9 mM	[47]
6R3K	2.20	JQT (BMS-1166)	1.85	[53]
6RPG	2.70	KDW	3.00	[54]
6VQN	2.49	R81	0.4	[55]
7BEA	2.45	TK2	16.80	[56]
7DY7	2.42	HOU	27.80	[57]
7NLD	2.30	UGZ	2.07	[53]
7VUN	2.00	8H7	8.90	[58]
8OR1	3.50	VYC	2.4	[59]
8K5N	2.20	I7M	1.8	[60]
8R6Q	2.17	WEW	<0.5	[61]

**Table 4 pharmaceuticals-17-00316-t004:** The three main hot regions and the corresponding hot spots on the PD-L1 surface detected by Holak and co-workers [44] analyzing the X-ray structure of the hPD-1/PD-L1 complex (PDB ID: 4ZQK).

Hot Region	Hot Spots	Note
1	Tyr56, Glu58, Arg113, Tyr123, Met115	Deepest cleft with predominantly hydrophobic character
2	Met115, Ala121, and Tyr123	Located near the previous hot region
3	Asp122, Tyr123, Lys124, Arg125	Extended groove with multiple H-bond donor and acceptor groups

**Table 5 pharmaceuticals-17-00316-t005:** Principal computational methods applied for the investigation of the hPD-1/PD-L1 binding mechanism.

Methods	Force Field	Thermodynamic Ensemble	Ref.
Classical and accelerated MD	AMBER FF14SB	NVT/NPT	[64]
Classical MD	AMBER03	NVT/NPT	[70]
Classical MD	CHARMM36	NPT	[65]
Classical MD	AMBER99SB-ILDN	NVT/NPT	[66]
Classical MD, MM/GBSA	AMBER FF14SB	NVT/NPT	[67]
Classical MD, MM/PBSA	OPLS/AA	NVT/NPT	[69]
Classical MD, infrequent MTD	AMBER03	NVT/NPT	[71]
Classical MD, MM/GBSA/IE	AMBER FF14SB	NVT/NPT	[68]

MD: molecular dynamics; MTD: metadynamics.

**Table 6 pharmaceuticals-17-00316-t006:** Key PD-L1 residues identified in different articles studying the binding with small molecules.

Studied Ligands	Computational Approach	PD-L1 Hot Spot Residues	Reference
29 BMS derivatives	Docking Binding free energy calculation	_A_Tyr56, _B_Tyr56, _B_Asp122, _B_Lys124, _B_Arg125	[72]
Several biphenyl derivatives from literature and patents	Docking	_A_Tyr56, _B_Tyr56, _B_Asp122	[73]
35 BMS derivatives	MD AS-IE	_A_Tyr123, _B_Tyr56, _B_Met155, _A_Met115, _B_Gln66, _A_Asp122	[75]
6 BMS derivatives	FMO/GRID-DRY	_A_Tyr56, _A_Met115, _A_Ala121, _A_Tyr123, and _B_Ile54, _B_Tyr56, _B_Met115, _B_Ala121	[80]
4 BMS derivatives	FMO 3D-SPIE	Tyr56, Glu58, Gln66, Met115, Asp122, Arg125	[81]

**Table 7 pharmaceuticals-17-00316-t007:** Principal features of the discussed virtual screening campaigns.

PDB ID	Database (# Compounds)	Program	Screening Protocol	Emerging PD-L1 Inhibitor	Activity (μM)	Ref
5J89	Specs (200,000)	Schrödinger Canva	docking study clustering	APBC	IC_50_ 27.82	[87]
5J89	natural compounds of ZINC12 (180,131)	iDock AuPosSOM Data Warrior	docking study filtering by iDock score (−9.95 kcal mol^−1^) Contact Fingerprint Analysis filtering by drug-likeness properties clustering (FragFP descriptor)	ZINC12529904	IC_50_ 0.1	[88]
* n.r.	ZINC database	ParDOCK Amber suite	docking study filtering by docking score (−8 kcal mol^−1^) MD simulations	Ponatinib	IC_50_ 1.91	[89]
5J89	NCI, Enamine, Specs, or in-house (900,000)	MOE FAF-Drugs4	docking-based 50 GA for rapid screening filtering by docking score re-docking 500 GA for top scoring compounds filtering by Lipinski’s rule of 5	69	IC_50_ 0.096	[90]
5J89	In-house (807) Life Chemicals (4994)	Schrödinger	Docking study (Asp122 &Lys124 hot spots) filtering by docking score	VIS310	Kd 8.13	[91]
5J89	ChEMBL25 (1.9 M)	AutoDock	filtering by Lipinski’s rule of 5 first docked once top-scored docked 5 times clustering by FCFP_4 fingerprints druggability	17	IC_50_ 0.0278	[57]
5J89	Specs (200,000)	Schrödinger	filtration by PAINS Pharmacophore model generation Docking study Clustering Induced fit docking	CBPA	Kd 48.10	[92]
5J89	Specs (540,807 s.m.) & DrugBank (1925 FDA-approved drug)	Schrödinger	3D pharmacophore model Docking study (3 steps: HTSV, SP, XP) Filtering by ADME and druggability	Raltitrexed Safinamide AK-968/40642641	Indirect in vitro experiments (↑ in immune cell proliferation)	[93]
5N2F, 5NIU, 6R3K, 5J89, 5J8O, 5N2D, 6NM8	10,000 approved or investigational drugs	AutoDock Vina (SB-VS) ROCS (LB-VS)	Ensemble docking (7 crystal structure) Visual inspection Filtering by shape similarity (7 crystal ligands) Docking study (top 1000 compounds)	Pyrvinium	IC_50_ 29.66	[94]
5N2F	bioactive molecules of Cayman Chemical database (16,191)	Open Drug Discovery Toolkit (ODDT) AutoDock Vina Desmond	Multiple 2D descriptors (FP1, FP2, Layered, MACCS, Morgan, RDKit) Random Forest models Docking study Molecular Dynamics	CRT5	IC_50_ 22.35 IC_50_ 33.65	[95]

* n.r. = not reported.

## Data Availability

Data sharing is not applicable.

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
