# Peer review of "A Comprehensive Computational Insight into the PD-L1 Binding to PD-1 and Small Molecules"

_pharmaceuticals, 2024, doi:10.3390/ph17030316_

Round 1

Reviewer 1 Report

Comments and Suggestions for Authors

Fantacuzzi et al. provide a comprehensive review of the new technological advances for targeting the PD-1/PD-L1 axis focusing mainly on the role of small molecules.

The paper is well-written and tries to cover the computational, chemical, and biological aspects of this interaction. However, a few changes and additions could improve the quality of their work and its readability.

Despite the authors stating their decision to avoid focusing on docking experiments, I think it is important to mention some work related to Alphafold predictions. In my opinion, especially in this moment in history where Artificial Intelligence (AI) is particularly relevant, it is worth spending a paragraph describing how AI (Alphafold and large language models (LLMs)) could help identify new small molecules that could target the PD-1/PD-L1 axis.

Some examples can be found here:

PD-L1 binders discovered by Alphafold:

https://www.ncbi.nlm.nih.gov/pmc/articles/PMC10628240/

Small molecules inhibiting CTLA-4 discovered through AI: https://www.nature.com/articles/s44276-023-00035-5

Review of Alphafold applications for immune checkpoint inhibitors:

https://www.mdpi.com/1422-0067/24/6/5908

On the other hand, I think some introductory information is missing, which would be useful for the reader to get a better feeling on the topic. For example, the introduction on Molecular Dynamics (MD) could be improved, adding some more basic knowledge on how the force field is used for the simulations. Similarly, several other technical terms are just mentioned, but not explored properly, such as the packages MOE, the Schrödinger Suite, the Partial Least Square Discriminant analysis, and the flexible docking studies. Some other terms weren’t defined in the text, such as per-residue-based decomposition, the grid Inhomogeneous Solvation Theory (GIST), and Hydration Site Analysis (HAS).

I feel that Section 4 is hard to read and follow. I found it too much descriptive and too little summarized, without a story that is followed through.

Some minor comments would be in line 290 to replace the term “principal” with “main”, and to move the citation 79 in line 474.

Finally, I think it’s useful to acknowledge that identifying the perfect small molecule doesn’t make it a drug and that many other additional steps should be included.

For example, the authors mention in the introduction that because of the poor outcome of immune checkpoint inhibitors, they have been used often in combination therapies. In this regard, some studies have tried to use computational approaches to predict whether combination therapy with immune checkpoint inhibitors would work or not in a certain cancer type, such as the following: https://www.mdpi.com/2076-3425/9/9/221

Comments on the Quality of English Language

None

Author Response

Fantacuzzi et al. provide a comprehensive review of the new technological advances for targeting the PD-1/PD-L1 axis focusing mainly on the role of small molecules.

The paper is well-written and tries to cover the computational, chemical, and biological aspects of this interaction. However, a few changes and additions could improve the quality of their work and its readability.

Answer: We would like to thank the reviewer for his/her insightful review of our manuscript and his/her useful suggestions, which can certainly help to improve the quality and impact of our work.

Despite the authors stating their decision to avoid focusing on docking experiments, I think it is important to mention some work related to Alphafold predictions. In my opinion, especially in this moment in history where Artificial Intelligence (AI) is particularly relevant, it is worth spending a paragraph describing how AI (Alphafold and large language models (LLMs)) could help identify new small molecules that could target the PD-1/PD-L1 axis.

Some examples can be found here:

PD-L1 binders discovered by Alphafold:

https://www.ncbi.nlm.nih.gov/pmc/articles/PMC10628240/

Small molecules inhibiting CTLA-4 discovered through AI: https://www.nature.com/articles/s44276-023-00035-5

Review of Alphafold applications for immune checkpoint inhibitors:

https://www.mdpi.com/1422-0067/24/6/5908

Answer: We followed the indication of the Reviewer and added a new paragraph 5 named “Application of AI-based methods and innovative computational approaches to study the immune-checkpoint inhibition.” where we discussed the application of AlphaFold to develop a new anti-PD-L1 antibody and deep convolutional neural networks to discover small molecules inhibiting CTLA-4 (page 23).

  1. Application of AI-based methods to the study of immune checkpoint inhibition

Artificial intelligence (AI) has had a major impact on technologies and all fields of science, including structural biology and in silico drug discovery studies. Apart from the already discussed VS campaign by Patil et al. [96], not many other studies report the application of these approaches to the discovery of PD-L1 ligands. Most of the literature mentioning PD-L1 and AI is based on the application of these methods to aid in the diagnosis and prediction of responses to AI treatments. [97]

AI is behind a breakthrough in structural biology represented by AlphaFold and similar software (RosettaFold, OpenFold, and ESMFold), which are able to predict the 3D structure of a protein from its amino acid sequence [98, 99]. This specific computational method was used to predict the anti-PD-L1 antibody and antigen structures of the humanized 3D5 antibody, named h3D5-hIgG1, as well as the structure of the corresponding h3D5/PD-L1 complex [100]. Experimental results showed that h3D5-hIgG1 is characterized by an extraordinary binding affinity to the PD-L1 protein compared to the parental murine 3D5 antibody. Other Alphafold applications aimed at identifying PD-L1 inhibitors have recently been reviewed by Sobral et al. [101]

The power of AI in drug discovery has also been applied to other targets in the immune checkpoint inhibitor space, as shown in a recent study using an AI algorithm based on deep convolutional neural networks to identify small-molecule inhibitors of cytotoxic T-lymphocyte-associated protein 4 (CTLA-4) that disrupt the CTLA-4/CD80 interaction [102]. This AI approach was used to virtually screen a library of 10 million compounds. The most promising compounds were evaluated using biochemical, biophysical, immunological, and animal assays to demonstrate their ability to inhibit the CTLA-4/CD80 pathway.

Although the application of AI-based methods to the identification of PD-L1 ligands has not been widely exploited, the increasing number of available tools and technical capabilities will certainly encourage the application of these methods to this target.

On the other hand, I think some introductory information is missing, which would be useful for the reader to get a better feeling on the topic. For example, the introduction on Molecular Dynamics (MD) could be improved, adding some more basic knowledge on how the force field is used for the simulations.

Answer: We agree with Reviewer that a more accurate description of classical MD and force field (FF) can be useful especially for non-expert readers. Therefore, we added the following new sentences to explain in more detail the MD protocol and the FF application in the initial part of paragraph 3:

  1. Computational studies contributing to the binding mechanism comprehension

Molecular dynamics (MD) is a powerful computational approach that allows exploring the conformational space of large biological systems.

More specifically, in a classical MD protocol, it is possible to calculate the force applied to each atom in the system and then use Newton's laws of motion to predict the spatial position of each atom as a function of time [62]. The resulting trajectory describes the conformational change of the 3D structure during the simulation time. The length of a trajectory is therefore a critical point of the simulation.

Forces in an MD simulation are calculated using a molecular mechanics force field (FF), which contains structural parameters (for example, the length of each covalent bond) as well as a set of equations to define electrostatic interactions between atoms and other types of interatomic non-covalent interactions [62]. The type of FF adopted for MD simulation can significantly influence the computational results and therefore should be selected carefully.

                                               …

Similarly, several other technical terms are just mentioned, but not explored properly, such as the packages MOE, the Schrödinger Suite, the Partial Least Square Discriminant analysis, and the flexible docking studies. Some other terms weren’t defined in the text, such as per-residue-based decomposition, the grid Inhomogeneous Solvation Theory (GIST), and Hydration Site Analysis (HAS).

Answer: We believe that a more detailed description of the theoretical aspects of all computational approaches (e.g., flexible docking, hydration site analysis, etc.) and specific software (e.g., Schrödinger Suite, MOE, etc.) mentioned in our manuscript is beyond the scope of this review.  Readers interested in delving deeper into the mentioned methodologies can certainly find comprehensive data in the literature.

I feel that Section 4 is hard to read and follow. I found it too much descriptive and too little summarized, without a story that is followed through.

Answer: We thank the Reviewer for his/her comment. Section 4 has been largely revised as suggested.

Some minor comments would be in line 290 to replace the term “principal” with “main”, and to move the citation 79 in line 474.

Answer: The suggested modifications have been reported in the manuscript.

Finally, I think it’s useful to acknowledge that identifying the perfect small molecule doesn’t make it a drug and that many other additional steps should be included.

For example, the authors mention in the introduction that because of the poor outcome of immune checkpoint inhibitors, they have been used often in combination therapies. In this regard, some studies have tried to use computational approaches to predict whether combination therapy with immune checkpoint inhibitors would work or not in a certain cancer type, such as the following: https://www.mdpi.com/2076-3425/9/9/221

Answer: We agree with the Reviewer’s observation and decided to insert a more extended discussion on this point in the Conclusions section (page 25).

In addition to the need to find a good ligand, one of the most challenging aspects of ICI is the limited number of patients who respond to therapy. Better profiling of cancer protein expression and the adoption of combination therapies may increase the number of patients who can benefit from immunotherapy. [103] Recently, a series of in silico methods were applied to simulate the different intracellular signaling affecting the PD-1/PD-L1 pathway in neuroblastoma (NBM) [104]. In particular, they developed a specific network of protein kinase cascades where the corresponding Michaelis–Menten kinetics parameters were used to create a system of ordinary differential equations. The resulting computational model represents an interesting tool to predict the relation between NBM tumor phenotype and the response of anti-PD-1/PD-L1 therapy as well as to manage immunotherapeutic treatment of NBM patients. This latter represents an example of the even more relevant role that computational studies can play in the valuable field of immune checkpoint modulation.

Reviewer 2 Report

Comments and Suggestions for Authors

Interesting and comprehensive review, with special congratulations for the author's initiative not to report "docking for the sake of docking" a posteriori studies of already available compounds, but to focus only on experimentally validated virtual screening!

My main concern with the review is that (to my taste) it is something not critical enough with the MD studies claiming "to explain the binding mechanism" - showing us structural details that cannot be read from experiment. True - but, as good old Niels Bohr one put it "if it cannot be measured, it does not exist", so I really wanted to have the author's final word on the following question: Did those studies really HELP someone in designing a new ligand, or were they just "post mortem" analysis of already known binders... concluding things that strictly apply to ... the already known binders! For, it is nice to highlight "free energy per residue" (whatever that means - I do not see how you can RIGOROUSLY "cut" the integral over the phase space, a.k.a "partition function", into contributions from residues - these are empirical scores of a certain utility... or not. Because, whatever those "free energy scores" will be, (a) they will simply give you the list of residues in direct contact with the ligand - I wonder whether there is enough sampling, even at 1 µs, to capture long-range allosteric effects, and I do not think these really concern the studied systems. Then, (b) whatever free energy map you compute... it will only apply to the ligand you have computed it with! Bring in a new ligand, with unexpected interaction modes, and your old map will be as helpful as a map of Borneo when you are travelling to Texas. So - I'd like the authors not to be shy from pointing out these fundamental limitations of all these "fancy" studies - for example, taking a QM-based method which (as the authors clearly state) is best to highlight charge-transfer/polar interactions... and then apply it to score the DRY (hydrophobic) probe? Does this not deserve a risen eyebrow in the text? And, most important - please highlight ANY direct positive impact of these MD studies, like "George carefully read the mechanistic hypotheses in paper P and came up with a novel very potent ligand". If there is no George... please say so!

Comments on the Quality of English Language

English is understandable, but rather suboptimal. For example, you do not prefix names by "the" - you do not say "The Charles is King of England", neither "the Bristol-Myers-Squibb..." There is no pharma company called "Merk" - there are two companies called "Merck", one in the US, one in Europe - but none connected to Angela Merkel. Another "pearl" is "The programmed death (PD)-1 was identified for the first time by Ishida" - I really doubt that Ishida was the first professional assassin in human history, but I rather suspect he identified the Programmed Death "protein". 

Author Response

Interesting and comprehensive review, with special congratulations for the author's initiative not to report "docking for the sake of docking" a posteriori studies of already available compounds, but to focus only on experimentally validated virtual screening!

My main concern with the review is that (to my taste) it is something not critical enough with the MD studies claiming "to explain the binding mechanism" - showing us structural details that cannot be read from experiment. True - but, as good old Niels Bohr one put it "if it cannot be measured, it does not exist", so I really wanted to have the author's final word on the following question: Did those studies really HELP someone in designing a new ligand, or were they just "post mortem" analysis of already known binders... concluding things that strictly apply to ... the already known binders!

For, it is nice to highlight "free energy per residue" (whatever that means - I do not see how you can RIGOROUSLY "cut" the integral over the phase space, a.k.a "partition function", into contributions from residues - these are empirical scores of a certain utility... or not. Because, whatever those "free energy scores" will be, (a) they will simply give you the list of residues in direct contact with the ligand - I wonder whether there is enough sampling, even at 1 µs, to capture long-range allosteric effects, and I do not think these really concern the studied systems. Then, (b) whatever free energy map you compute... it will only apply to the ligand you have computed it with! Bring in a new ligand, with unexpected interaction modes, and your old map will be as helpful as a map of Borneo when you are travelling to Texas. So - I'd like the authors not to be shy from pointing out these fundamental limitations of all these "fancy" studies - for example, taking a QM-based method which (as the authors clearly state) is best to highlight charge-transfer/polar interactions... and then apply it to score the DRY (hydrophobic) probe?

Does this not deserve a risen eyebrow in the text? And, most important - please highlight ANY direct positive impact of these MD studies, like "George carefully read the mechanistic hypotheses in paper P and came up with a novel very potent ligand". If there is no George... please say so!

Answer: We would like to thank the Reviewer for his/her appreciation of our review. The Reviewer’s considerations are certainly pertinent, but answering his/her question is not trivial. In most of the studies, reported results tend to complete the experimental observations with a quantitative or dynamic description. As these data mostly overlap with experimental data, it is hard to elicit the effective exploitation of these data by researchers looking for new ligands.

Concerning the computational methods explored, we understand the Reviewer’s comments, but the evaluation of largely applied methods, such as the per residue energy decomposition, or FMO is not in the aim of the present review.

Some comments have been added in the Conclusions section (page 24).

Most investigations in this field use the available X-ray data to perform docking, MD simulations, and other structure-based approaches. The studies presented here largely confirm the involvement of PD-L1 residues in both the binding of BMS derivatives and the dimerization process, as observed experimentally by Holak. The identification of hot-spot residues in the PD-L1 binding region for both PD-1 and small molecules converged on hydrophobic contacts with Tyr56, Met115, Ala121, and Tyr123, as well as electrostatic contacts with Asp122, Lys124, and Arg125. Most of the computational studies completed the binding profile of the studied ligands, therefore, since they mostly overlap with experimental data, it is not trivial to elicit the effective contribution of these methods to the identification of new small-molecule binders.  

English is understandable, but rather suboptimal. For example, you do not prefix names by "the" - you do not say "The Charles is King of England", neither "the Bristol-Myers-Squibb..." There is no pharma company called "Merk" - there are two companies called "Merck", one in the US, one in Europe - but none connected to Angela Merkel. Another "pearl" is "The programmed death (PD)-1 was identified for the first time by Ishida" - I really doubt that Ishida was the first professional assassin in human history, but I rather suspect he identified the Programmed Death "protein". 

Answer: all the manuscript has been carefully revised to improve English.

Reviewer 3 Report

Comments and Suggestions for Authors

PD-1/PD-L1 interaction has been a hot target in cancer therapy. Computational studies have delved into the binding mechanism of small molecules to PD-L1 and the dimerization process, showcasing the role of structure-based approaches. This review summarizes articles exploring computational analyses of PD-L1 binding with PD-1 and small molecule ligands, emphasizing the growing significance of these techniques in chemical biology and drug discovery.

In the “Introduction” part, the authors clearly presented the role of PD-1/PD-L1 pathway in immune silencing and cancer development, summarized currently approved mAbs with their limitations, and the discovery and potential of small molecule binders.

In section 2, the authors mainly reviewed the crystal structures of PD-L1, PD-1/PD-L1 complex, and PD-L1/small molecule complexes.

In section 3, the authors summarized the computer-aided methods for studying the binding mechanism of PD-L1 to PD-1/small molecules and its dimerization induced by ligands. Hot spots residues on PD-L1 and key chemical features in PD-L1 binders were described.

Section 4 highlighted computational studies contributing to the discovery of new ligand for PD-L1 via virtual screening.

The manuscript is well-organized and comprehensively reviewed computational studies on PD-L1 binding mechanism, which is valuable for researchers in this field. Following are suggestions to strengthen the impact of the manuscript:

1.     Line 152: I think the author refers to “Table 2” not “Table 1”

2.     Line 193: Could the author specify which domain is interacting with PD-1?

3.     Figure 5: Could the author add the small molecule BMS-202 in Figure 5A? Also, it’s better to add a panel to show the dimerization of PD-L1 in the presence of ligand.

Author Response

PD-1/PD-L1 interaction has been a hot target in cancer therapy. Computational studies have delved into the binding mechanism of small molecules to PD-L1 and the dimerization process, showcasing the role of structure-based approaches. This review summarizes articles exploring computational analyses of PD-L1 binding with PD-1 and small molecule ligands, emphasizing the growing significance of these techniques in chemical biology and drug discovery.

In the “Introduction” part, the authors clearly presented the role of PD-1/PD-L1 pathway in immune silencing and cancer development, summarized currently approved mAbs with their limitations, and the discovery and potential of small molecule binders.

In section 2, the authors mainly reviewed the crystal structures of PD-L1, PD-1/PD-L1 complex, and PD-L1/small molecule complexes.

In section 3, the authors summarized the computer-aided methods for studying the binding mechanism of PD-L1 to PD-1/small molecules and its dimerization induced by ligands. Hot spots residues on PD-L1 and key chemical features in PD-L1 binders were described.

Section 4 highlighted computational studies contributing to the discovery of new ligand for PD-L1 via virtual screening.

The manuscript is well-organized and comprehensively reviewed computational studies on PD-L1 binding mechanism, which is valuable for researchers in this field. Following are suggestions to strengthen the impact of the manuscript:

  1. Line 152: I think the author refers to “Table 2” not “Table 1”.
  2. Line 193: Could the author specify which domain is interacting with PD-1?
  3. Figure 5: Could the author add the small molecule BMS-202 in Figure 5A? Also, it’s better to add a panel to show the dimerization of PD-L1 in the presence of ligand.

Answer: We thank the Reviewer for his/her valuable comments.

We have answered all questions, in particular:

  1. The table number in the text has been carefully checked and corrected (Table 3).
  2. In paragraph 2, line 191, we have added the following sentence: "The distal Ig-type V-like domain is responsible for interaction with PD-1 and small molecules".
  3. We have decided to remove the ligand structure from the figure for clarity. In fact, the purpose of the figure is to highlight the different arrangement of PD-L1 residues when it binds PD-1 and           BMS-202. A panel A has been added to Figure 5 showing the dimeric form of PD-L1 in complex with BMS-202.
